# The Study of Growth and Performance in Local Chicken Breeds and Varieties: A Review of Methods and Scientific Transference

**DOI:** 10.3390/ani11092492

**Published:** 2021-08-25

**Authors:** Antonio González Ariza, Ander Arando Arbulu, Francisco Javier Navas González, Sergio Nogales Baena, Juan Vicente Delgado Bermejo, María Esperanza Camacho Vallejo

**Affiliations:** 1Department of Genetics, Faculty of Veterinary Medicine, University of Cordoba, 14014 Cordoba, Spain; angoarvet@outlook.es (A.G.A.); anderarando@hotmail.com (A.A.A.); sergionogalesbaena@gmail.com (S.N.B.); juanviagr218@gmail.com (J.V.D.B.); 2Animal Breeding Consulting, S.L., University of Cordoba, 14014 Cordoba, Spain; 3Institute of Agricultural Research and Training (IFAPA), 14004 Córdoba, Spain; mariae.camacho@juntadeandalucia.es

**Keywords:** native breeds and varieties, nonlinear modelling, growth curves, poultry

## Abstract

**Simple Summary:**

The present review evaluates twenty years (2001 to 2021) of the study of growth and performance in local chicken breeds worldwide. The assessment of methodological approaches and their constraints when intending to fit for data derived from often endangered autochthonous populations was performed. The evaluation of conditioning factors on the impact that publications reporting on research progresses in the field have on the scientific community and how such advances are valued suggests the need to seek new methodological alternatives or statistical strategies. Such strategies must meet the requirements of local populations which are characterized by reduced censuses, a lack of data structure, highly skewed sex ratios, and a large interbreed and variety variability. The sustainable conservation of these populations cannot be approached if scientific knowledge on their productive behaviour is not reinforced in a manner that allows distinctive products to be put on the market and be competitive.

**Abstract:**

A review of the scientific advances in the study of the growth and performance in native chicken breeds and varieties over the past 20 years was performed. Understanding the growth patterns of native breeds can only be achieved if the constraints characterizing these populations are considered and treated accordingly. Contextually, the determination of researchers to use the same research methods and study designs applied in international commercial poultry populations conditions the accuracy of the model, variability capturing ability, and the observational or predictive performance when the data of the local population are fitted. Highly skewed sex ratios favouring females, an inappropriate census imbalance compensation and a lack of population structure render models that are regularly deemed effective as invalid to issue solid and sound conclusions. The wider the breed diversity is in a country, the higher the scientific attention paid to these populations. A detailed discussion of the most appropriate models and underlying reasons for their suitability and the reasons preventing the use of others in these populations is provided. Furthermore, the factors conditioning the scientific reception and impact of related publications used to transfer these results to the broad scientific public were evaluated to serve as guidance for the maximization of the success and dissemination of local breed information.

## 1. Introduction

Chicken breeds make up the majority of all avian breeds in the world (63%). Halfway through February 2021, out of the 875 chicken breeds officially recognised in Europe, 10.64% were extinct and 41.16% were considered to be at risk and included in the “vulnerable” and “critical” classifications according to DAD-IS (Domestic Animal Diversity Information System) FAO database [1]. Moreover, the average number of gaps in ex situ collections of selected crop gene pools and the proportion of local breeds classified as being at risk out of all the breeds whose risk of extinction is known to have been quantifiably developed by the FAO Commission on Genetic Resources for Food and Agriculture in the following sustainable development goals: (2.5.1.) the number of plant and animal genetic resources for food and agriculture secured in either medium or long-term conservation facilities and (2.5.2.) the proportion of local breeds classified as being at risk, not-at-risk or at unknown level of risk of extinction [2]. Only 8.58% of the total European chicken breeds are considered not to be at risk, while for 36.50% of European chicken breeds, not enough information in regard to their status was available, hence they were classified as unknown (Figure 1). 

The worldwide number of hens outnumbers the worldwide human population by a ratio of 2.5 to 1. Of the almost 17,000 million birds, approximately half are concentrated in Asia and a quarter in Latin America and the Caribbean. Europe and the Caucasus comprise more than 13% of the worldwide chicken population, followed by Africa with 7%.

As can be inferred from these numbers, indigenous or local breeds represent most of the worldwide poultry genetic diversity. These breeds are classified depending on whether they are registered in a single country (native), in several countries in the same region (regional cross-border), or several regions (international cross-border). The percentages for each of these categories may vary considerably from region to region [3]. As suggested in Figure 1, population data are frequently missing (36.50% unknown status), making risk assessment extremely difficult. The lack of data is a consequence of the difficulties that the monitoring of small livestock populations involves as a direct consequence of the weak attention that most governments generally pay to poultry despite their pivotal roles in livestock food security, rural livelihoods, and gender equity [4,5].

The loss of native breeds not only represents a severe threat from the perspective of the disappearance of genetic resources, but also simultaneously translates into the irreversible loss of social, cultural, and inheritance resources. These breeds are an integral part of the evolutionary diversity of each region [6]. Furthermore, it is important to highlight the competitive advantages that the concept of autochthonous breeds indirectly generates for livestock farmers as beneficiaries of the different rural development policies. Breed conservation is the most efficient way to preserve biodiversity [7].

Perhaps the most relevant driving element of this recent drastic loss in poultry genetic resources is the development of productively competitive hybrid strains associated with mergers of breeding companies and the global consolidation of commercial poultry farms [8]. This event has also translated into significant losses in experimental lines, most of which occur in research centres, given the increasing difficulty to find the necessary funds for the conservation of these resources [9].

The need to produce food at the lowest cost has increased the census of highly productive foreign domestic breeds at the cost of displacing native breeds [10]. The process of extinction of breeds not only ends up with the irreparable disappearance of genetic resources but also weakens the populations as a side consequence of genetic erosion as a result of the separated or combined effect of ineffective selection programs on small population sizes [11].

The resilience of local poultry breeds and their ability to thrive in the framework of sustainable systems while the outcomes of production farming practices are maximized ensures the consolidation of these resources [12]. However, it is not their potential as a productive alternative but the possibilities that local breeds offer to obtain differentiated and unique products, whose properties may not only significantly differ from the products obtained through the exploitation of commercial lines, but also which may cover a wider spectrum of consumer needs and thus may target a more specialized market [13].

The enhancement of the commercial opportunities of local products may be one of the most efficient strategies for the conservation of local genotypes and this is the point where the circular economy cycle closes. Product differentiation ensures the satisfaction of particular niches in a rather suitable manner than standardized products, given the ascription of products to the local breeds from which they derive, and the area in which these were produced confers them with an added value which in most of the cases may be supported by a chemical, organoleptic, or even a cultural heritage and traditional basis or a combination of all [13,14]. 

The proper development of these strategies can only be achieved if products and the animals which produce them are thoroughly known. Local chicken breeds’ productive applications could be sorted into three main purposes: meat production, egg production, and aesthetics [13]. According to the report by Shahbandeh [15], the projected global consumption of poultry meat will amount to 151.83 metric kilotons by 2030 from the 133.35 metric kilotons expected for 2021 (carcass weight equivalent), which represents an increase of around 13.86%. This global situation provides evidence of the relevance of meat performance and growth as breeding criteria.

Contextually, growth can be defined as the weight gain of the animal until it reaches adult size. This growth accelerates during the early stages of the individual’s life; therefore, there is a greater weight gain when the animal approaches adulthood, so that, when developing the growth curve, there is a line ascending sigmoid curve. As the individual reaches its adult size, the growth rate is altered, and therefore there is a change in curvature. It is at this point (inflection point) where the highest growth rate is identified. From this point on, growth gradually slows down and the growth rate slows down. This is where growth stabilizes, creating a continuous trend, which mathematically coincides with a horizontal asymptote [16]. The growth can be fixed in some coordinates of weight and time employing a series of points, obtaining growth curves. They can be summarized into several biologically interpretable parameters and provide estimates of growth rate and weight at maturity [17].

Some authors have reported the fact that the smaller the breed, the faster they mature [18]. Indeed, meat production requires birds to be ready to butcher in 4 months weighing more than two pounds live weight. As opposed to local meat chicken breeds, commercially used breeds have been reported to experiment with a sharp and quick growth, while they may not reach the quantity or quality of meat that markets are currently demanding at a rather higher cost which is not maintained if production conditions are not standardized. 

In this context, most of the local chicken breeds and varieties have valuable genetic features and constitute efficient profitable resources that could provide valuable breeding material for the poultry industry worldwide. However, the lack of apparent competitiveness has prompted these resources to be severely outlooked through the years, which has been translated in census reduction and circumscription to specific world areas which makes difficult the implementation and deems incorrect the application of those study methods that are regularly applied in commercial poultry-related science.

For instance, the regular mathematical functions which are normally applied to commercial poultry lines to reveal growth patterns, determine potential cause and effect relationships, and develop productive strategies may no longer respond to the properties and limitations of the data derived from local breeds [19].

Therefore, the present review first aims to evaluate the international scope and framework of the study of growth and performance in local chicken breeds. Second, the determination of the methods used to evaluate growth patterns in these genetic resources and which constraints these methods may face due to the limitations (availability, gender ratio imbalance, among others) is approached to build a guide that may facilitate the design of future studies involving local chicken genetic resources which in most of the cases are endangered and scarce.

## 2. Review of Data Collection and Analysis

### 2.1. Data Collection

The present study was carried out following the methodology previously described by McLean and Navas González [20] and Iglesias et al. [21]. Two independent repositories were used to obtain the data from the present study: www.sciencedirect.com and www.scholar.google.es (accessed on 16 July 2021) [22]. The decision related to the inclusion of the aforementioned repositories was made based on the fact that they comprise tools that enable data extraction for analysis in a way that other platforms such as https://www.ncbi.nlm.nih.gov/pubmed/ do not, as suggested by Iglesias et al. [21] and Gehanno [22].

For the search, we used the subsequent keywords: mathematical/nonlinear/non-linear growth models and followed each one with the words native/indigenous/local poultry or chicken breed or any related term in their semantic fields [23]. The data were collected during June 2021 to ensure the publications included in the present review were updated. Only the documents that compared non-linear models for growth performance in native chicken breeds were retained. The selected papers were included in a database, which comprised individual registries for each article. Each record comprised the variables sorted into six variable clusters. The first cluster comprised the variables linked to the population under study (breed and variety); the second cluster comprised those factors related to the location of the study (country and continent); the third cluster comprised the method-related factors such as the growth model and the number of parameters; the fourth cluster was linked to the study design properties (male and female number, total sample, female and male observations, and total observations). The fifth cluster related to model performance (goodness of fit and flexibility criteria) and comprised the variables of the determination coefficient (R^2^), the mean squared error (MSE), the root mean squared error (RMSE), the residual standard deviation (RSD), the Akaike Information Criterion (AIC), and the Bayesian Information Criterion (BIC), while the sixth and last cluster comprised variables related to Scientific Impact such as the year of publication, the Journal, the Indexation status, the Impact factor quartile, and the database in which the publications were eventually published. The nature, maximum, minimum, and levels of the variables included in the analysis are summarized in Table 1.

### 2.2. Data Analysis

#### 2.2.1. Assumption Testing

The Shapiro–Francia W’ test (for 50 < *n* < 2500 samples) was used to discard gross violations of the normality assumption in the dependent variables considered in the study. The Shapiro–Francia W’ test was performed using the sfrancia routine of the test and distribution graphics package of the Stata Version 16.0 software (College Station, TX, USA). The rest of the parametric assumptions (Levene’s and Mauchly’s W tests and the Tolerance and Variance Inflation Factor) were performed using SPSS Statistics for Windows, Version 25.0, IBM Corp (2017).

#### 2.2.2. Statistical Approach Decision

As the parametric assumptions were not met (*p* < 0.05), the use of nonparametric approaches to analyze the data were chosen. Consequently, the monotonic relationship (whether linear or not) among the continuous variable pairs (Table 1) was tested through the Spearman correlation coefficient using the Bivariate routine of the Correlate procedure of SPSS Statistics for Windows, Version 25.0, IBM Corp (2017). The Kruskal–Wallis H, Dunn, and Independent median tests were performed to detect differences in the distribution and median across the breeds and varieties. The association between the nominal variables was measured through Cramér’s V. According to Cohen [24], one of the most accurate interpretations of this parameter depends on the degrees of freedom as presented in Table 2. A frequency analysis was run to determine the likelihood of the model being used across breeds and varieties. A frequency analysis was tested using the Frequencies routine of the Descriptive Statistics procedure of SPSS Statistics for Windows, Version 25.0, IBM Corp (2017).

## 3. Growth and Performance Modelling

### 3.1. Models Used in the Literature to Fit for Growth and Performance

The evaluation of the literature resources revealed the use of a total of twenty models to study the growth patterns of native poultry breeds. The growth functions can be sorted into three categories as suggested by Darmani Kuhi, et al. [25]: those which only represent a decreasing returns profile (for instance, monomolecular, exponential with sharp cut-off), those describing a smooth sigmoid profile with a fixed inflection point (for instance, Gompertz, logistic), and those characterized by a sigmoid profile with a flexible inflection point (for instance, von Bertalanffy, Richards). Table 3 presents the SPSS model syntax for each of the 20 models found. This SPSS model syntax was ready to be copied and pasted in the non-linear regression task from the Regression procedure of SPSS version 25.0. Additionally, the references in which the use of each model was reported are also enclosed.

Figure 2 reports that the most frequently used models to describe the growth performance of native breeds are Gompertz, Logistic, and Richards models. The exponential nature of the functions of these models has been deemed the main reason for the improved fitting ability of the aforementioned methods [25]. Gompertz and Von Bertalanffy’s models are the most frequently used models to fit for growth in local genotypes. On the other hand, acceptable results have been reported after the use of models such as the Brody model which has traditionally been used to fit for growth in larger species as it does not tend to overestimate the weight in light poultry species [16,54]. 

Studies in which the Gompertz–Laird and Brody models were used suggest there is a need to use a higher number of males in the observational sample size since these models are so sensitive to the imbalance of the number of individuals in both sexes [55]. 

The consideration of flexible growth functions as an alternative to the simpler equations (with a fixed point of inflection) to describe the evolution of body weight in time is recommended given they are easy to fit and provide a closer fit to data points (flexibility). They therefore have smaller values for MSE, RMSE, RSD, and RSS than computationally and parametrically simpler models. The addition of an extra parameter has been reported to be an effective alternative in those cases in which no clue is present about the behavior of a particular data sets [55].

### 3.2. Goodness-of-Fit and Flexibility Criteria

R^2^ measures the ability of a model to capture the variability for a certain trait in a population. The Kruskal–Wallis H test revealed differences in the distribution of determination criterion (R^2^) across breeds and varieties (*p* < 0.05). Intra-breed or inter-variety homogeneity may parallelly translate into lower explicative and predictive errors, thus leading to an improved model fitting accuracy. By contrast, native avian breeds usually are heterogeneous populations that need a higher number of individuals in the sample to obtain acceptable values in the goodness-of-fit and flexibility criteria. A high variability in the data makes it compulsory for models to account for high flexibility, otherwise, the performance in the characterization of biological growth curves of these genotypes decays.

Additionally, the correct characterization of growth and performance in a population (already defined breeds or not) must be carried out using a balanced number of weights from both sexes to prevent the incorrect application and interpretation of statistical data analysis. Contrastingly, the evaluation of the literature references highlighted a remarkable trend of using a greater number of females than males in the studies. In this regard, researchers attempt to compensate for the low experimental sample sizes by increasing the observational samples through the number of females. In this manner, although the implementation of this strategy efficiently causes an increase in R^2^, the likelihood of a Type I error increases as well, translating into an overinflation of variability which is captured and thus measured by R^2^ as a direct consequence. This lack of accuracy was also denoted when the same test suggested that MSE should be conditioned by the breed and the variety of chickens. 

A sex ratio imbalance was found in almost all the reviewed papers. In this regard, the variance overinflation probably derived from the fact that highly sex imbalanced populations were being modelled comprising both sexes altogether. The literature has suggested that the aptitude to which specific poultry genotypes are destined may condition the use of a greater number of animals of one sex or another [56]. 

Therefore, in breeds destined for egg production there will be a large number of females which in turn translates into larger observational samples, while in breeds that present a meat orientation, the number of males and the observations that derive from them is consequently larger. 

Simultaneously, the endangerment status of the population has also been reported to condition sex ratio imbalances as the number of females in breeds in conservation status must be well defined to preserve the breed at the same time that we prevent the effects derived from inbreeding depression [57]. 

As aforementioned, the statistical nature of the data set derived from native poultry populations and the statistical limitations of using R^2^ make it necessary to evaluate this criterion with caution. For instance, R^2^ does not show whether the estimates and predictions of the coefficients are biased, which is the reason why residual plots must be examined. Furthermore, the R^2^ does not indicate whether a regression model is adequate or not, and this means R^2^ can be low in a suitable model or R^2^ can be high in an incorrect data fitting model, as it is strongly dependent on the number of observations. 

Researchers often attempt to improve the outcome of statistical parameters by increasing the number of observations; however, the R^2^ value can decrease as a consequence of a higher number of outliers and therefore the sample noise in highly variable populations [58]. Therefore, the use of R^2^ is appropriate as long as it is accompanied by other model selection criteria.

In the last few years, authors have tried to reduce the models and even produce certain variations in each of them so that they fit specific biological curves [59]. Additionally, the development of some flexibility criteria, such as corrected Akaike’s information (AICc) or BIC, have been aimed at penalizing models with a high number of parameters in their formula [60]. 

It was observed that the authors did not usually use the flexibility criteria AIC and BIC. These criteria are based upon concepts of entropy and information by focusing on a statistical approach. While AIC provides a relative estimate of the missing information when a particular model is used to represent the process that generates the data, BIC is based on the probability function [61]. When biological growth model simulations were performed with a very low number of animals, AIC (observational/explicative) worked extremely well and showed a better yield and performance in comparison to BIC (predictive) [60]. As a result, the computation of AIC and BIC, among other flexibility criteria, has proven to be essential in the selection of the best fit model in native breeds, for which the sample size is usually small.

As reported in recent studies [10] for growth characterization and following the methodology described by Van Vleck [62], the inclusion of a combined selection index (ICO) is appropriate to sort models depending on their better fit and flexibility properties, since goodness-of-fit and flexibility criteria may differ in terms of their most desirable values and their magnitude. The use of this index allows the position in the rank for each of the goodness-of-fit and flexibility criteria determined for each model to be summarized.

### 3.3. Constraints and Particularities for Growth Modelling in Native Genotypes (Breeds and Varieties)

Non-linear models have been widely contrasted as suitable methods to fit for growth in native poultry [18,50]. However, other alternatives, such as mixed models with random and fixed effects have been formerly suggested in the literature to fit for the same aim [63]. Several desirable properties can be found using mixed models (random and fixed). For instance, the fact that nonlinear mixed model coefficients allow a stochastic prediction of covariates such as the mean age that birds need to achieve certain body weight and its variation, allows for unique new decision-support modelling applications. In turn, as suggested by Afrouziyeh et al. [63] these methods could be used in stochastic modelling to evaluate the economic impact of management decisions in poultry breeding-related industries. However, their use could be conditioned by the nature of the data derived from the study of local populations, given such data may not meet certain assumptions [64]. 

In this context, as suggested by previous authors, mixed models need at least five levels or groups for a random intercept term to achieve robust estimates of variance [65]. Other authors [66,67] have suggested that fixed or random effects that have lower than five levels may perform an inaccurate estimation between population variance, and due to variance estimates could reach values near to zero, which could be derived in a model similar to non-linear modelling [66] or be non-zero, but this is incorrect when the small number of levels from which samples were used is not representative of the true distribution of means. This could suppose a variance and covariance distortion and consequently, this low number of levels could result in a fixed or random effect [68].

In maximum restricted likelihood methods (for other methods such as those based upon the Bayes theorem), sex is normally considered to be a fixed effect, since among other reasons, this factor accounts for an *a priori* number of already know possibilities or levels (i.e., males and females). In this sense, the randomization of sex may lead to model degeneracy, a biased estimation of the random effect variance, an inaccurate estimation of the random effect variance, and a high error potential for questions related to random effects [69]. 

The randomization of other factors such as the individual itself (animal permanent environmental effect), when animals are repeatedly measured along the course of the study, implies that the random effects corresponding to the same animal are correlated. Repeated measurements are a cornerstone in growth and performance evaluation studies. Consequently, this acts against the fulfilment of the assumption of independence of observations. Furthermore, random mixed models can be unstable when sample sizes across groups are highly unbalanced, which is likely to occur in native poultry populations, in which male/female ratios are frequently highly skewed in favour of higher female numbers.

Finally, an incorrect parameterization of the model’s random effects could yield unreliable model estimates. Among others, failing to identify dependency structures that meet the assumption of non-independence and failing and testing the significance of fixed effects at the wrong level may eventually lead to pseudo replication and inflated Type I error rates or a rejection of a true null hypothesis (“false positive” finding or conclusion) which could be prevented through the use of residual degrees of freedom for fixed effects [70].

The fitness of a factor as a fixed effect may easily provide a statement of the significance variation (differences across sexes, breeds, or varieties). However, in these cases, the separate evaluation and comparison of the same factor has been deemed effective in preventing the aforementioned situations from occurring. It is in these contexts in which non-linear models may be preferable when fitting growth data belonging to local breeds, especially in those cases in which the aforementioned constraints occur. Nonlinear models differ in terms of their computational complexity (the mathematical operations that they involve) and parametrical complexity (the number of curve shape parameters that they include). This is supported as the number of parameters considered in a model and the nature of its mathematical function may indeed be responsible for the better fitting properties of certain models against others, as suggested by Pizarro et al. [71,72], which becomes especially important in data-limited contexts.

Parallelly, Bayesian inference approaches, which randomize all factors, could be considered a feasible alternative and maybe a preferable method to test for differences among levels of random effects when drastic unbalanced level data limitations have hindered the robust application of frequentist statistical analyses [73]. 

In this context, a strong model–breed association was observed (Cramér’s V = 0.529; *p* < 0.001) which, however, was not significant when the relationship between variety and model was tested (*p* = 0.562). This suggests the choice of some growth models over others may depend on the breed rather than the variety being fitted. The rationale for this may rely on the fact that inter-breed variability may be broader than inter-variety variability; hence, certain models may indeed better fit for the biological growth curve of some breeds in question, but no differences may be found if these same breeds diversify into varieties. This finding has been ascribed, to a large extent, to the productive application of breeds, if these are breeds that produce eggs, meat, or are dual-purpose breeds [19,74]. In this context, a direct relationship between the maturity weight and the relative growth rate has been reported in the literature. In particular, some authors have suggested that there is a high probability that large-format breeds are less precocious than the smallest and lightest [75].

The endangerment situation that breeds face worldwide indirectly conditions the size of experimental (number of animals) and observational samples (number of observations per animal). Native poultry breeds are characterized by highly skewed populations in which the female/male ratio favours one of the sexes, in a manner that is normally linked to the purpose the animals in the population are used for (egg, meat, or dual-purpose). Frequently, males represent smaller numbers in the population, thus they act as a source from which a limited number of observations can be obtained [18,50]. 

In this context, researchers are frequently compelled to use a comparatively larger number of females (which are still limited) than males in studies. As a response to this sex ratio imbalance, the most common trend found was the increase in the number of observations (larger observational samples from limited experimental samples), which in turn ends up with the imbalance properties of the sample growing.

In these situations of a high imbalance, the minority class is often poorly represented and lacks a clear structure [55]. This has been reported to directly hinder the robustness of modelling methods and the correct application of statistical approaches [76]. In this regard, the use of randomized methods has been deemed inadvisable due to a high potential variance induced by the imbalance ratio [55]. Other methods which can empower the minority class and predict or reconstruct a potential class structure seem to be a promising direction. 

The decomposition of the original problem into a set of subproblems, for instance, modelling sexes separately with each group being characterized by a reduced imbalance ratio, has been suggested as an alternative to counteract these statistical obstacles [76].

## 4. Scientific Transference

### 4.1. Year of Publication

A trend in journals to publish studies using simpler models to fit for growth using larger experimental and observational samples from local chicken breeds over the years has been reported. This denotes the effort of researchers to adapt the nature of the local breeds to the requirements of journals. In this context, journals are one of the most relevant elements in the conservation chain of these local genetic resources. Chicken breeds often comprise endangered animal populations with limited censuses, and which therefore lost the attention that they normally received from the broad public (breeders lost their interest due to not receiving sustainable income) or administration (subsidies no longer covered production expenses). The loss of attention from owners and authorities also brought about the lack of attention from research entities, which left these populations with poor opportunities to thrive. In this context, market and scientific visibility and consumer knowledge of these local breeds is essential for their conservation, as the process seeking their official recognition must be complexly supported not only by research but also by the protection of a societal background (cultural heritage, productive sustainability, market profitability, product distinctiveness, among others) [13,77].

### 4.2. Study Georeferencing (Continents and Countries)

Figure 3 presents the distribution of studies across countries, with Nigeria (in Africa) and Spain, and Italy (in Europe) being the most active countries in terms of research publications basing upon the study of local populations. A very strong association was reported between the impact factor, quartile, and database of the journal and the continent or country where the study was carried out (*p* < 0.001). When working with native breeds, there are a large number of limitations regarding the availability of animals and even infrastructures where the research is carried out, due to the low budget that local breeds have in many countries compared to the economic resources that are conferred to rather productive commercial strains [9,78]. Institutional support is necessary to develop investigation studies in relation to local breeds. Hybrid commercial strains and other foreign breeds have the financing of big poultry multinational integrators based in countries such as China or the USA (countries sharing genetic connections based upon historical market relationships) which also translates into higher scientific attention being paid to them [79]. All in all, even if a balance of the territorial distribution of growth studies of local breeds across the different continents is shown (Figure 3), not all countries will manage to obtain data of sufficient quality as to be of interest for the most elite journals.

### 4.3. Method and Study Design-Related Research Impact Conditioning Factors

The results derived from the Kruskal–Wallis H test suggest that the use of certain models is associated with publications in a higher quartile, and therefore, a higher impact factor (*p* < 0.05). The lack of novelty of the models seemed to penalize in terms of highly impacted journal publication. In this regard, it was proved that Gompertz and Von Bertalanffy’s models were associated with studies published in journals with lower impact factors. These models are two of the three most used models in the fitness of growth in local chicken genotypes and frequently appear in the scientific scene. Editorial boards of journals highly value the novelty in the approaches followed as studies must always seek efficient alternatives which provide more accurate results at a lower simplicity cost.

In contrast, journals with higher impact factors had a trend to publish studies in which goodness-of-fit and flexibility criteria reported close to the reference values, but which used simple models with a low number of parameters in their formula. In this context, the Brody model has frequently been associated with studies published in journals with higher impact factors. Studies reporting a high R^2^ could indistinctly be observed in low and high impact factor journals, although a trend to find articles with a large R^2^ in less relevant journals was observed. This finding may be supported by the fact that large values of R^2^ are linked to highly imbalanced limited samples (either experimental or observational), which translate into the misfunction of the fitting ability of models. In this regard, the increase in the number of individuals considered for studies simultaneously caused a decrease in variability overinflation, and therefore R^2^ derived from the reduction in the likelihood of Type I errors [80].

## 5. Conclusions

Although the models by Gompertz and Von Bertalanffy widely cover the scientific scene for growth modelling-related research, a trend in highly impacted indexed journals to explore statistical parametrically simpler but computationally more complex alternatives progressively has occurred. A high sex ratio imbalance may strongly limit the statistical approaches that can be solidly implemented, although the current methods to counteract this situation may not be effective enough (increasing number of females to compensate the lack of male observations). The use of mixed models including breed, variety, or sex as either random or fixed factors has been prevented in favor of nonlinear models given the first may not respond to the distribution properties of the data derived from endangered autochthonous populations. Productive application strongly conditions the better fitting and flexibility performance of models. Growth pattern variability differences between breeds and varieties promotes the fact that a wider scope of models is needed to respond to the existing biological growth patterns. Countries accounting for higher levels of local poultry breed diversity may play a leading scientific role in the dissemination of knowledge related to these local populations and a higher consciousness among breeders, authorities, and research entities may occur.

## Figures and Tables

**Figure 1 animals-11-02492-f001:**
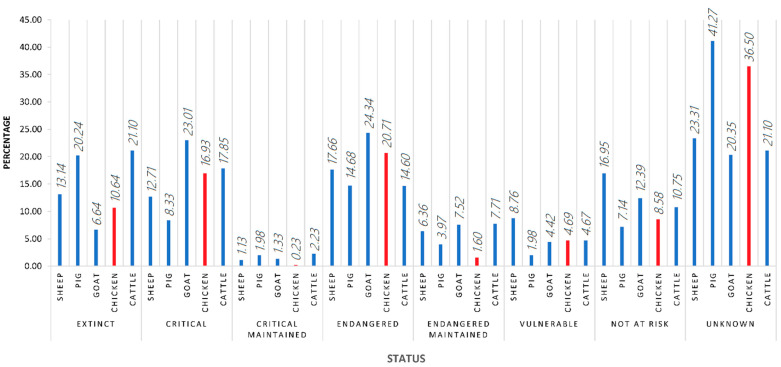
The classification status of European chicken breeds according to FAO DAD-IS halfway through February 2021.

**Figure 2 animals-11-02492-f002:**
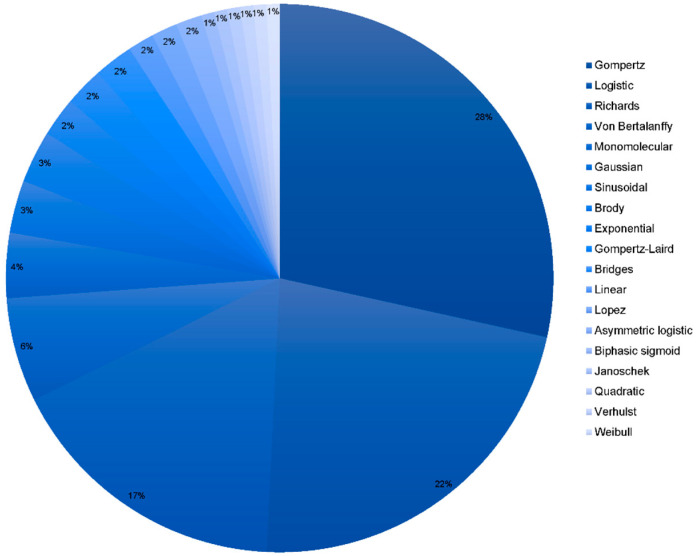
Frequency analysis of the models used to fit for growth and performance in native breeds. The darker the blue in the cheese graphic, the most frequently the model was used. The model in legend appears in decreasing frequency order.

**Figure 3 animals-11-02492-f003:**
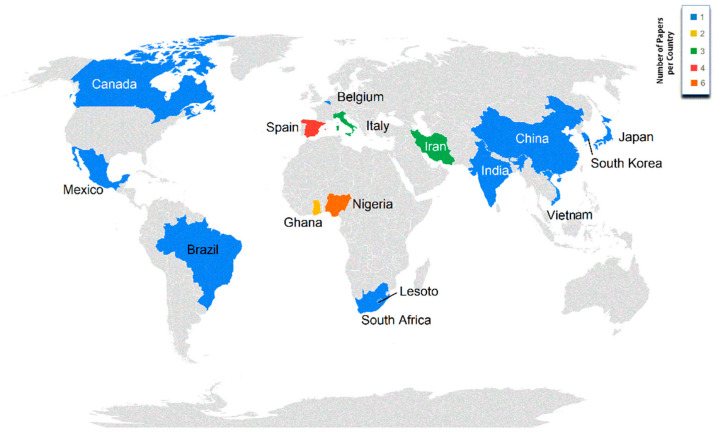
Territorial distribution and number of papers per country.

**Table 1 animals-11-02492-t001:** Nature, maximum, minimum (numeric and ordinal variables), and levels (nominal and ordinal variables) of the variables considered in the study.

Variable	Variable Set	Type	Levels (Maximum–Minimum)
Breed	Population	Nominal	41 breeds
Variety		Nominal	69 varieties
Country	Study Georeferencing	Nominal	16 countries
Continent		Nominal	Africa, Asia, Europe, America, and Australasia
Growth model	Method	Nominal	20 models (see Table 2 for model definition)
Number of model parameters		Numeric	2 to 6 model parameters
Male/Female sample	Study design	Numeric	11 to 749 males/12 to 1255 females
Total sample		Numeric	17 to 2004 individuals
Total male/female observations		Numeric	85 to 16,000 males/80 to 31,808 females
Total observations		Numeric	170 to 47,808 observations
R^2^ (variance explicative potential)	Goodness of fit and flexibility criteria	Numeric	0.01 to 1 for males/0.16 to 1 for females
MSE (model accuracy)		Numeric	1443 to 37,596,433 for males/1107 to 39,687 for females
RMSE (model accuracy)		Numeric	0.03 to 128 for males and 7.17 to 106 for females
RSD (deviation from the theoretical model)		Numeric	11,47 to 197 for males/10.41 to 191 for females
AIC (observative ability)		Numeric	49.42 to 74,719 for males/44.21 to 21,142 for females
BIC (predictive ability)		Numeric	60.12 to 74,739 for males/54.15 to 94,595 for females
Year of publication	Scientific impact	Ordinal	2002 to 2020
Journal		Nominal	24 journals
Indexed		Nominal	Yes, no, not at the moment of data collection
Impact factor		Numeric	0.14 to 2.217
Quartile		Ordinal	Q1, Q2, Q3, Q4
Data Base		Nominal	Not indexed, JRC, SJR, Scopus

**Table 2 animals-11-02492-t002:** Degrees of Freedom dependent interpretations for Cramér’s V.

Interpretation	No Effect	Effect Is Not Presumed but Can Be Detected with Additional Laboratory Techniques	Effect Is Presumed and Can Be Detected but Additional Laboratory Techniques Are Needed	Effect Can Be Detected with the Naked Eye
Degress of Freedom (df)	Negligible	Small	Medium	Large
1	0.00 < 0.10	0.10 < 0.30	0.30 < 0.50	0.50 or more
2	0.00 < 0.07	0.07 < 0.21	0.21 < 0.35	0.35 or more
3	0.00 < 0.06	0.06 < 0.17	0.17 < 0.29	0.29 or more
4	0.00 < 0.05	0.05 < 0.15	0.15 < 0.25	0.25 or more
5 or more	0.00 < 0.05	0.05 < 0.13	0.13 < 0.22	0.22 or more

**Table 3 animals-11-02492-t003:** SPSS Model syntax of mathematical models.

Model	SPSS Model Syntax	References
Asymmetric logistic	b0/((1 + b1*EXP(-b2*t))**(1/b3))	[26]
Biphasic sigmoid	b0/1 + EXP(b1*(b2-t)) + (b3/(1 + EXP(b4*(b5-t)))	[27]
Bridges	b0 + b1*(1-EXP(-(b2*t **b3)))	[28,29]
Brody	b0*(1-b1*EXP(-b2*t))	[18,29,30]
Exponential	b0*(1 + b1)*t	[31]
Gaussian	b0*(1-b2*EXP(-b1*t**2))	[32]
Gompertz	b0*EXP(-b1*EXP(-b2*t))	[18,26,28,29,30,31,32,33,34,35,36,37,38,39,40,41,42,43,44,45,46,47,48]
Gompertz–Laird	b0*EXP((b1/b2)*(1-EXP(-b2*t)))	[49,50,51]
Janoschek	b0-(b0-b1)*EXP(-b2*(t**b3))	[29]
Linear	b0 + b1*t	[36,52]
Logistic	b0*(1 + EXP(-b2*t))**(-b3)	[18,26,28,29,30,31,32,33,35,36,38,40,41,42,43,44,45,46,48,50]
Lopez	(b0*b1*b2 + b3*t*b2)/(b1*b2 + t*b2)	[33,35]
Monomolecular	b0*(1-b1*EXP(-b2*t))	[31,39]
Quadratic	b0 + b1*t + b2*t**2 + b3	[52]
Richards	b0*(1-b1*EXP(-b2*t))**b3	[26,28,29,30,32,33,35,36,38,39,41,43,44,48,50,53]
Sinusoidal	b0*(1-b1*COS(b2*t + b3))	[32]
Verhulst	b0/(1 + b1*EXP(-b2*t))	[18]
Von Bertalanffy	b0*(1-b1*EXP(-b2*t))**3	[18,30,33,40,44,45,46,50]
Weibull	b0-(b1*(EXP(-b2*(t**b3))))	[35]

t: age in days.

## Data Availability

Data will be made accessible from corresponding authors upon reasonable request.

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
