# Peer review of "The Study of Growth and Performance in Local Chicken Breeds and Varieties: A Review of Methods and Scientific Transference"

_animals, 2021, doi:10.3390/ani11092492_

Round 1

Reviewer 1 Report

Dear authors,

I would like to report that it was a pleasure to me reading your manuscript. Although the work is based on a literature review, it presents a relevant objective which is reached with a well-organized set of Tables, figures and text. I have no suggestions or recommendations to do about the manuscript content. All positive aspects were reported to Editor in order to give him/her the as better as possible support to take an editorial decision.

Just the following brief comments/questions…you have selected some articles but why you did not include the articles of meat quality of these breeds in the study? Are there no studies reported in literature about it? Is the absence of these articles due to the scope and the objective of the study which proposed to characterize the native breeds and populations? My comment is based on my personal perception.

Congratulations to the authors for the good work!

Author Response

I would like to report that it was a pleasure to me reading your manuscript. Although the work is based on a literature review, it presents a relevant objective which is reached with a well-organized set of Tables, figures and text. I have no suggestions or recommendations to do about the manuscript content. All positive aspects were reported to Editor in order to give him/her the as better as possible support to take an editorial decision.

Response: We thank the reviewer for his/her kind comments.

Just the following brief comments/questions…you have selected some articles but why you did not include the articles of meat quality of these breeds in the study? Are there no studies reported in literature about it? Is the absence of these articles due to the scope and the objective of the study which proposed to characterize the native breeds and populations? My comment is based on my personal perception.

Response: Meat Quality was first considered to be a section of the paper but the statistical approach and the nature and number of the articles found made it compulsory to prepare a different paper. Otherwise, the complexity and the extension of the paper would have been considerably increased, which would have made approaching the topic unproperly. The other paper is almost finished and will be submitted in the fore coming weeks.

Congratulations to the authors for the good work!

Response: We thank the reviewer for his/her kind comments again.

Reviewer 2 Report

I found the review very interesting with a particular intriguing approach to the conservation of local chicken genetic resources.

I just would like to suggest you some minor corrections.

Please, enlarge the columns in tables 1 and 2, because are difficult to reading. If it is possible, please separate each “variable set” in table 1 with a line.

Figure 2: please, increase the font size of the percentage values.

Line 233-239: the sentence is too long, please reformulate.

Please add to the reference Mosca et al., 2015 https://doi.org/10.3382/japr/pfv044

Author Response

Comments and Suggestions for Authors

I found the review very interesting with a particular intriguing approach to the conservation of local chicken genetic resources.

Response: We thank the reviewer for his/her kind comments.

I just would like to suggest you some minor corrections.

Please, enlarge the columns in tables 1 and 2, because are difficult to reading. If it is possible, please separate each “variable set” in table 1 with a line.

Response: Changed.

Figure 2: please, increase the font size of the percentage values.

Response: Changed.

Line 233-239: the sentence is too long, please reformulate.

Response: Changed.

Please add to the reference Mosca et al., 2015 https://doi.org/10.3382/japr/pfv044

Response: Added.

Reviewer 3 Report

This review article provides important insights into a degradation in the approaches being used to characterise local chicken breeds.

It is recommended that the authors consider making reference to the Sustainable Development Goal targets 2.5.1 and 2.5.2.  Also worthy of consideration is monitoring different carcass components that are valued across different cultures, including the giblets.

The article would be improved by the input of an editor fluent in English prior to re-submission.

The article will provide a useful addition to the literature once a number of issues have been addressed as outlined above and in the attached file. 

Author Response

Comments to the authors

This review article provides important insights into a degradation in the approaches being used to characterise local chicken breeds.

Response: We thank the reviewer for his/her kind comments.

It is recommended that the authors consider making reference to the Sustainable Development Goal targets 2.5.1 and 2.5.2.

Response: Added.

Also worthy of consideration is monitoring different carcass components that are valued across different cultures, including the giblets.

Response: Meat Quality and carcass composition was first considered to be a section of the paper but the statistical approach and the nature and number of the articles found made it compulsory to prepare a different paper. Otherwise, the complexity and the extension of the paper would have been considerably increased, which would have made approaching the topic unproperly. The other paper is almost finished and will be submitted in the fore coming weeks.

The article would be improved by the input of an editor fluent in English prior to re-submission.

Response: The manuscript was revised by a Cambridge University ESOL examinations to improve manuscript readability and to correct typos and grammar incongruencies.

The article will provide a useful addition to the literature once a number of issues have been addressed as outlined above and below.

Response: We thank the reviewer for his/her kind comments.

Line(s) Comments

14 Consider changing ‘native’ to ‘local’ or ‘indigenous’

Response: Changed.

15 Change ‘constrains’ to ‘constraints’

Response: Changed.

54 Consider changing ‘individuals’ to ‘birds’ for clarity; otherwise some readers could confuse this with humans

Response: Changed.

75 Did you mean ‘driving’ rather than ‘motor’?

Response: Changed.

262 Did you mean ‘reviewed’ rather than ‘revised’?

Response: Changed.

282 Researchers?

Response: Changed.

309 Constraints?

Response: Changed.